# Violence against women on Twitter in India: Testing a taxonomy for online misogyny and measuring its prevalence during COVID-19

Nabamallika Dehingia[1]*, Julian McAuley[2], Lotus McDougal[3], Elizabeth Reed[4], Jay G. Silverman[3], Lianne Urada[5], Anita Raj[3]

1 UNICEF, UNICEF HQ, New York, New York, United States of America, 2 Department of Computer Science, School of Engineering, University of California San Diego, San Diego, California, United States of America, 3 Center on Gender Equity and Health, Department of Medicine, University of California San Diego, San Diego, California, United States of America, 4 Division of Health Promotion and Behavioral Science, School of Public Health, San Diego State University, San Diego, California, United States of America, 5 School of Social Work, San Diego State University, San Diego, California, United States of America

* ndehingia@unicef.org

**Data Availability Statement:** All relevant data necessary for replicating the study findings can be found here: https://doi.org/10.7910/DVN/H3IXDN.

## Abstract

### Background

Online misogyny is a violation of women's digital rights. Empirical studies on this topic are however lacking, particularly in low- and middle- income countries. The current study aimed to estimate whether prevalence of online misogyny on Twitter in India changed since the pandemic.

### Methods

Based on prior theoretical work, we defined online misogyny as consisting of six overlapping forms: sexist abuses, sexual objectification, threatening to physically or sexually harm women, asserting women's inferiority, justifying violence against women, and dismissing feminist efforts. Qualitative analysis of a small subset of tweets posted from India (40,672 tweets) substantiated this definition and taxonomy for online misogyny. Supervised machine learning models were used to predict the status of misogyny across a corpus of 30 million tweets posted from India between 2018 and 2021. Next, interrupted time series analysis examined changes in online misogyny prevalence, before and during COVID-19.

### Results

Qualitative assessment showed that online misogyny in India existed most in the form of sexual objectification and sexist abusive content, which demeans women and shames them for their presumed sexual activity. Around 2% of overall tweets posted from India between 2018 and 2021 included some form of misogynistic content. The absolute volume as well as proportion of misogynistic tweets showed significant increasing trends after the onset of COVID-19, relative to trends prior to the pandemic.

**Funding:** This study was funded by the Bill and Melinda Gates Foundation Grant No. OPP1163682. The funding body had no role in the design of the study and collection, analysis, and interpretation of data and in writing the manuscript.

**Competing interests:** The authors have declared that no competing interests exist.

## Conclusion

Findings highlight increasing gender inequalities on Twitter since the pandemic. Aggressive and hateful tweets that target women attempt to reinforce traditional gender norms, especially those relating to idealized sexual behavior and framing of women as sexual beings. There is an urgent need for future research and development of interventions to make digital spaces gender equitable and welcoming to women.

## Introduction

Violence against women (VAW) on online platforms is a violation of women's digital rights. It can push women out of online spaces, and impact their social, economic, as well as health outcomes [1]. One of the most prevalent forms of online VAW, online misogyny or hate speech against women refers to any content that tries to threaten, intimidate, and shame women, or any rhetoric that emphasizes the authority of men over women [1]. According to a multi-country online survey in 2019, around two-thirds of women users of the internet reported having received some form of sexist or hateful language designed to attack or humiliate them [2]. Such experiences can often cause increased anxiety, depression, and lower self-esteem among women [3, 4]. The extent and frequency of misogynistic attacks varies across different groups of women; younger women, and women belonging to marginalized racial or sexual identity groups are more at risk of experiencing online misogyny [2, 5–9]. The past two years have thus noted an increasing number of calls, for hate speech regulation on social media platforms [10, 11].

The majority of the existing research on online violence, including online misogyny, has focused on high-income countries, primarily due to greater digital access and use of social media platforms in these regions [12]. To our knowledge, no academic study has examined the characteristics of online misogyny in low- and middle-income countries (LMIC). This presents a key gap in literature, given that the use of social media, and consequently, the amount of online hateful content, has increased rapidly in LMICs in recent years [13, 14]. India, in particular, is an important geography to examine online misogyny. Compared to many other LMICs, adverse gender norms and offline forms of gender-based violence are more prevalent in the country [15, 16], putting women at a higher risk of experiencing violence in online spaces. Use of social media platforms is also on the rise in the country. As of January 2022, India had over 25 million active Twitter users, with a 30% increase in average daily users in 2020. Although Twitter has an urban bias; most Twitter users in the country are likely to be educated, living in cities, and of middle to high wealth status [17]. Nonetheless, Twitter is being frequently used as a key platform for feminist discussions and organizing in India [18], which can increase the likelihood of misogynist expression as a backlash [19]. The current study aims to contribute to the growing literature on online VAW on social media platforms, by measuring online misogyny on Twitter in India.

Limited global research suggests a potential increase in this form of violence since the pandemic, impacting millions of women daily [10]. An analysis of tweets and Facebook posts from South and South-east Asia by United Nations Women found 168 percent increase in misogynistic content during March-June 2020, when compared to the same period in 2019 [20]. However, despite many such analyses of the effect of COVID-19 on online content, little is known about the features and prevalence of online misogyny, and how it might have changed over the course of the pandemic. In this study, we take a first step to provide temporal

estimates of online misogyny on Twitter in India from 2018 to 2021, and systematically examine any changes in its rates of increase/decrease before and during COVID-19.

As early as May 2020, the United Nations Secretary General made a global appeal to tackle COVID-19 influenced hate speech on digital platforms, with hateful content covering "*stereotyping, stigmatization, and the use of derogatory, misogynistic, racist, xenophobic, Islamophobic or antisemitic language*" [21]. Potential drivers of such hate speech during the pandemic include increased isolation due to stay-at-home orders, greater use of social media platforms, greater exposure to polarizing and differing views on the social media platforms, and fear, uncertainty, and anxiety of living through the pandemic [22]. Isolation is a strong motivational factor for hate speech on social media [23], and the confined living conditions, health and financial worries following the lockdown can create tension and stress, increasing the risk for online expression of hate speech. With regards to hate speech against women in particular, global evidence in support of increasing cases of gender-based violence during the pandemic further points to a potential increase in violence against women on online platforms [24]. An analysis of Twitter data before and after the onset of COVID-19 will allow us to test this hypothesis of increased levels of misogyny on Twitter since the pandemic.

## Defining online misogyny

A key reason for the limited empirical evidence on online misogyny, is the lack of a standard taxonomy [25]. A majority of existing research on this topic has lacked theoretical considerations, classifying it as content that includes identifiable sexist slurs [26–28]. Earlier works on misogyny detection relied on lexicon-based methods, where misogynistic tweets were identified based on the presence of gendered abusive words [29, 30]. Anzovino et. al (2018) introduced a benchmark dataset of tweets classified as misogynistic, with online misogyny defined as consisting of five forms: a) slurring, b) stereotypes, c) sexual harassment and threats, d) dominance to preserve men's control, and e) derailing to justify VAW [31, 32]. Multiple other studies have used this definition for misogyny detection [33, 34]. We build on this taxonomy, by adapting it to the context of India, and locating it within a theoretical understanding of misogyny as a notion central to the feminist theory.

Feminist theory posits that misogyny, whether online or offline, stems from patriarchal values justifying men's control over women via degradation and violence; misogyny actively seeks to silence women and maintain the status quo of patriarchal gender roles [35]. Sobieraj describes intimidating, shaming, and discrediting as the three key strategies often adopted by perpetrators of online misogyny, where attackers draw on women's fear of sexual assault and physical violence to intimidate them, sexually objectify to publicly shame them, and discredit or dismiss their achievements as well as their fight to equality [1]. Asserting the inferiority of women, dismissing feminist movements, and objectifying women or viewing women primarily as an object of men's desire, are thus at the heart of misogyny [36]. Our preliminary research on misogyny detection in South-Asian countries noted an emphasis on dismissal of feminist thought and discrediting of women's claims of gender-based violence [37]. Guided by these theoretical and empirical works, we define online misogyny as any rhetoric or content that uses hostile and malicious language targeted at women, objectifies them, threatens them with physical or sexual harm, tries to assert their inferiority, justifies gender-based violence, or discredits feminist activists and their efforts. By qualitatively analyzing a large dataset of tweets (over 40,000 tweets), we first test whether this definition and taxonomy is valid for content posted on Twitter from India. This is followed by estimation of temporal online misogyny, using machine learning techniques that rely on the validated taxonomy. Our study objectives are two-fold: a) testing a taxonomy for online misogyny in India that is rooted in feminist

theory, and b) using this taxonomy, estimate whether prevalence of online misogyny on Indian Twitter changed since the pandemic, across a sample of 30 million tweets posted between 2018 and 2021. Given the current lack of research on the extent and forms of misogyny on Twitter in LMICs, our study hopes to generate evidence that can be used to advocate for policies and strategies to address misogyny on Twitter.

## Methods

### Online misogyny taxonomy

We define online misogyny as consisting of six broad and overlapping forms: a) sexist abusive content, b) sexual objectification, c) threatening to physically or sexually harm women (presence of sexist language that indicates the threat being directed at women), d) asserting inferiority, e) justifying VAW, and f) dismissing feminist efforts. We provide definitions and example tweets for each form of online misogyny as Supporting Information. This taxonomy is guided by feminist theoretical work on misogyny [36], prior research on online misogyny detection, particularly the studies by Anzovino et. al (2018), and our preliminary research on measuring online misogyny in South Asian countries.

### Extracting geolocated tweets

We used geotagged English language tweets posted from India between January 2018 and December 2021 (30 million tweets). All tweets were extracted using the official Twitter API for academic research, which allows extraction of historical tweets. The most recent 800–1000 tweets published every hour from January $1^{st}$ 00:00 2018 to December $31^{st}$ 23:00 2021 were collected.

We used supervised machine learning models to classify tweets as misogynist/non-misogynist, as well as the six different forms of misogyny. Fig 1 depicts the different steps involved in our analysis. We describe each step in detail in the following sections.

### Qualitative analysis of tweets for misogyny

Supervised machine learning models require a ground truth dataset. To that end, two trained undergraduate students first qualitatively coded a small subset of tweets, as misogynist/non-misogynist, and the six forms of misogyny. The selection of the subset of tweets was done using two approaches. First, from the large corpus of geotagged tweets between 2018 and 2020, we extracted posts containing representative keywords frequently used to harass and abuse women. To identify keywords, we contacted authors of an academic study on hate speech detection on Twitter. They used a list of keywords for different types of hate speech on Twitter, including misogyny [38]. We added a few words relevant to the Indian context to this list. Identified words included abusive terms, as well as generic words related to gender such as "*woman*", "*feminism*" etc. It is key to note here that there may be certain tweets that contain abusive words, but might be non-offensive in their use and overall meaning. The inclusion of all such tweets that contained abusive words allowed us to build a ground truth dataset that identified the nuances in the use of abusive words. With this process, we selected a subset of 35,672 tweets.

Next, we chose a random subset of 5,000 tweets from the large corpus of geotagged tweets from 2018–2020. This dataset was included to ensure that tweets unrelated to women or gender were also represented in our ground truth sample. The main goal of using the two separate approaches to build the ground truth dataset was to collect a set of tweets that would have an adequate representation of both misogynist and non-misogynist tweets. A total of 40,672 tweets were thus selected for qualitative coding, with these two processes.

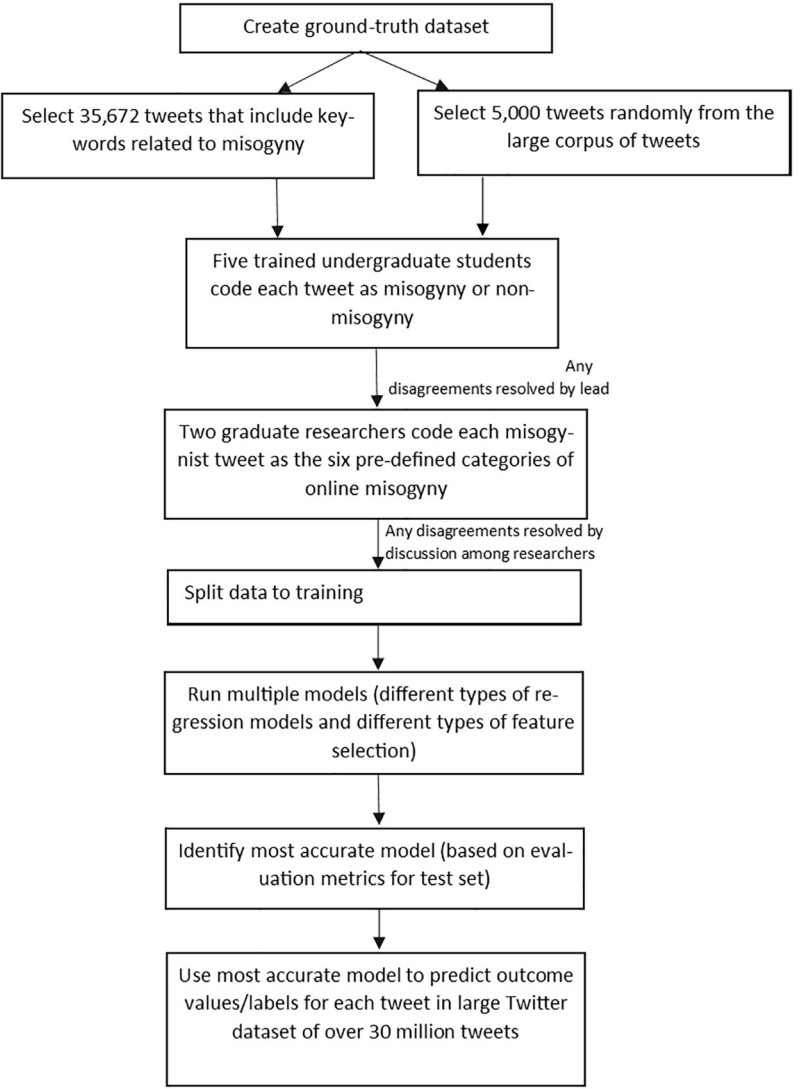

**Fig 1. Flowchart of processes involved in misogyny detection among tweets.**

The qualitative coding of the tweets was carried out in two phases. First, the subset of tweets was coded as misogynist or non-misogynist by five trained undergraduate students. Every tweet was classified by at least two coders. Inter-rater kappa scores were calculated to examine reliability of the coding [39]. Any disagreement between the coders was resolved by the lead author of the study. Next, all tweets classified as misogynist were coded by two graduate students experienced in gender research, as the six forms of online misogyny. Inter-rater kappa scores were calculated for this step as well. The qualitative coding of misogynist tweets provided us with the opportunity to validate our taxonomy of misogyny. The two researchers did not identify any content that was outside the purview of the six predefined categories.

### Selection of features or independent variables

Tweets were cleaned before running any machine learning model. This included deleting urls and ignoring punctuation and case. We also removed stop words, which are commonly used

words such as "is", "the", "a" etc., from our text data. The decision to drop stop words was made after running multiple models with and without stop words. Next, the data was processed with *text vectorization*, in order for the machine learning regression models to mathematically interpret it. Text vectorization refers to the process of creating features or input variables in a model (independent variables in public health terminology). There are multiple methods for generating features or independent variables from text data. We used three types of feature selection methods: a) term frequency-inverse document frequency (TF-IDF) unigrams, b) TF-IDF unigrams and bigrams, and c) TF-IDF unigrams, bigrams, and linguistic features (number of words and number of adjectives). These methods were chosen based on guidelines suggested by a previous study on abuse detection [32]. TF-IDF is weighted word frequency, and reflects how important a word is to a document in a collection or corpus [40]. While TF-IDF with unigrams concerns with single words only, TF-IDF with unigrams and bigrams includes sequences of two words along with singular words. In addition to weighted word frequencies, linguistic features such as length of the tweet or number of adjectives in a tweet can also provide meaningful information about the content of the tweet. It is possible that tweets that are misogynist are shorter in length and have a relatively higher number of adjectives (abusive words or sentiments).

## Running supervised machine learning models and predicting misogyny: Train and test models for best fit

We randomly split our data into a train and test dataset in a 70:30 ratio. The training dataset is used to train the machine learning model, while the test dataset is used to assess the performance of the trained model. To estimate the hyperparameters to be specified in the training model, we used a k-fold validation process. In this method, the training data are partitioned into k subsets of approximately equal size and one of the subsets becomes the validation set where the hyperparameters are validated. The remaining k-1 subsets are used as training data and this process is repeated.

We ran five different types of supervised machine learning models to classify tweets as misogynist/non-misogynist: a) Naïve Bayes (NB), b) Support Vector Machine (SVM), c) Ridge logistic regression, d) Multi-layer Perceptron Neural Network (MPNN), and e) Bidirectional Encoder Representations from Transformers (BERT). Naïve Bayes is a probabilistic model based on Bayes theorem and considers a strong independence assumption [41]. SVMs are based on a structural risk minimization principle, which aims to test a hypothesis for which we can guarantee the lowest true error. SVMs aim to create a hyperplane (*maximum-marginal hyperplane*), which separates the categories of the outcome label [42]. MPNN are complex non-linear models with a hierarchical or multi-layered structure. These models have been commonly used in text classification tasks [32]. BERT is a pre-trained language model (BERT is pretrained on a large corpus of English texts from Wikipedia and BookCorpus), with transformer architecture [43]. Transformers learn contextual relations between words; they are deep learning models where each output element is connected to every input element and the weightings between them are dynamically calculated. While previous models examine text sequences in a single direction (left-to-right or right-to-left training), BERT's key difference and innovation is the application of bidirectional training. The BERT Transformer encoder reads the entire sequence of words at once. Specifically, we used the *bert-large-uncased* version, which consists of 24 layers (1024 hidden dimensions), 16 attention heads, and a total of 340M parameters. The *Transformers* library in Python was used to implement this approach.

The performance of each model was evaluated on the test set, by comparing actual labels or outcome variable values with the values predicted by the trained model. We used two

evaluation metrics: (a) F1 score and (b) receiver operating characteristic—area under the curve (AUC). The receiver operating characteristic captures precision and recall at all potential decision thresholds, so the area under it is an appropriate metric to measure overall model performance. F1 score is the harmonic mean of precision and recall. The models with the best F1 and AUC estimates were chosen to predict the labels/outcome variable values for each tweet in our original dataset of over 30 million geotagged tweets posted between January 2018 to December 2021.

The analysis was repeated with each of the six forms of online misogyny as the outcome variable or label, with the sub-sample of tweets that was classified misogynist (N = 3894).

### Analysis of misogyny prevalence pre and during COVID-19

To assess whether misogyny prevalence increased post the onset of COVID-19, we used Interrupted Time Series Analysis (ITSA), a quasi-experimental design that can evaluate an intervention effect using longitudinal data. It is a segmented regression model, with different intercept and slope coefficients for the pre- and post-intervention time periods. Here, the intervention is on March 25, 2020, which is when the first nation-wide lockdown was imposed in India. The following equation specifies the ITSA model for our study:

$$Y_t = \beta_0 + \beta_1(T) + \beta_2(X_t) + \beta_3(X_tT)$$

$Y_t$ is the proportion of online misogyny related tweets, T is the time since COVID-19 related lockdown has been implemented in India (in days), $X_t$ is the dummy variable representing the imposition of lockdown, and $X_tT$ is an interaction term. $\beta_1$ represents the underlying pre-COVID-19 trend in misogyny prevalence, or the change in misogynistic tweets for every one unit time increase. $\beta_2$ captures the immediate effect of the onset of COVID-19, and $\beta_3$ represents the slope change in the proportion of misogynistic tweets following COVID-19, in comparison to trends prior to COVID-19. We ran interrupted time series regression models with two separate outcomes: a) total number of misogynistic tweets and b) proportion of misogynistic tweets per 100000 tweets.

## Results

### Qualitative assessment of misogyny across tweets

Around 8% of the tweets that were qualitatively analyzed were classified as misogynist (N = 40,672). Among those identified as misogynist, 57% were related to sexual objectification, 34% covered sexist abusive content, 6% justified violence against women, and 5% were content that dismissed feminism or feminist efforts. A small proportion of misogynist tweets included threats of harm (1%) and assertion of authority (1%). The two researchers who conducted the qualitative coding did not identify any content that was outside the purview of the six predefined categories of online misogyny. The coding had good reliability, with an inter-rater kappa score of 0.87.

The tweets classified as sexual objectification (57%) included text that sexualized women's body, shamed women for presumed sexual activity (*slut shaming*), and text that included lewd remarks and sexual solicitation. These tweets emphasized the treatment of women as sexual objects of men's sexual desire. The category "justify violence against women" (6%) included tweets that highlighted the prevalence of rape myth acceptance and victim blaming among Twitter users. Tweets aggressively asserted that women needed to act a certain way to reduce the risk of rape and violence (e.g., not wear short clothes, not go out at night). This category also included tweets that aimed to derail the conversation from VAW, by focusing on "fake

rape cases" in India. Around 10% of these tweets included the hashtag "#mentoo", with tweets noting men's victimization with fake rape allegations. Additionally, there were a small number of tweets (n = 3) that dismissed marital rape. For example, a tweet stated that all married women were free to be raped by their husbands.

The category on dismissal of feminist efforts included a number of tweets declaring feminism as a "virus", with 13% of the tweets including the word "feminazis". Around 16% of the tweets classified as dismissal of feminist efforts were also categorized as justifying VAW.

## Machine learning models to predict misogyny

We ran the different types of machine learning models, separately for each outcome variable: misogyny, sexual objectification, sexist abusive content, justifying VAW, dismissing feminist efforts, threatening to harm, and asserting authority. For the analysis with misogyny as the outcome variable, BERT performed the best [Table 1]. This was followed by the logistic regression models (with TF-IDF unigrams). The Naïve Bayes had the lowest values of AUC as well as F1 score, for misogyny prediction.

## Predicting online misogyny across tweets from 2018–2021 and examining prevalence

We used the specifications from our best performing model to predict misogyny across the large Twitter dataset of 30 million tweets. Each tweet in this dataset was classified as misogyny or non-misogyny by this model. Overall, 1.6% of the geotagged tweets collected from India included misogynist content.

We ran separate interrupted time series models with total number of misogynistic tweets, and proportion of misogynistic tweets per 100000 tweets as the outcomes (Table 2). The volume of misogynistic tweets increased significantly by 30 times immediately after the onset of COVID-19, with increasing trends in the following months between April 2020-December 2021. The proportion of misogynistic tweets per 100000 tweets also showed significant and sustained increasing trends after the onset of COVID-19, relative to trends prior to the pandemic. However, there was a significant decrease immediately after the onset of COVID-19, followed by the increasing trends.

## Discussion

Digital social media portals are often signaled to be democratic public spaces. However, we find that similar to the offline world, they are home to increasing gender inequalities. Our study observes online misogyny on Twitter in India to be prevalent, with increasing trends

**Table 1. Evaluation of different types of machine learning models for misogyny prediction.**

| | Support Vector Machine | | Naïve Bayes | | Logistic Regression | | Multi-layer Perceptron Neural Network | | BERT | |
|---|---|---|---|---|---|---|---|---|---|---|
| **Misogyny as outcome label** | | | | | | | | | | |
| | AUC | F1 | AUC | F1 | AUC | F1 | AUC | F1 | AUC | F1 |
| TF-IDF unigrams | 95.68 | 70.20 | 73.74 | 39.07 | 96.78 | 70.39 | 95.00 | 62.38 | - | - |
| TF-IDF unigrams and bigrams | 95.25 | 69.39 | 74.04 | 47.44 | 96.76 | 69.99 | 96.01 | 64.74 | - | - |
| All features (TF-IDF unigrams, bigrams, and linguistic- number of words and adjectives) | 93.42 | 64.96 | 74.09 | 39.62 | 96.68 | 69.99 | 94.11 | 61.38 | - | - |
| Feature engineering with transformers | - | - | - | - | - | - | - | - | **90.82** | **84.00** |

**Table 2. Interrupted time series analysis to examine trends in misogyny prevalence pre and during COVID-19.**

| | Outcome: Number of tweets classified as misogynistic | Outcome: Proportion of tweets per 100000 tweets classified as misogynistic |
|---|---|---|
| | β (95% CI) | β (95% CI) |
| Time since COVID-19 onset (β indicates underlying pre-COVID-19 trend) | -0.05 (-0.06- -0.02) *** | -0.03 (-0.11–0.05) |
| Intervention (β indicates level change immediately following COVID-19) | 30.85 (16.25–45.45) *** | -215.64 (-275.29- -155.98) *** |
| Interaction term between time and intervention (β indicates slope change, or sustained change following COVID-19) | 0.09 (0.05–0.13) *** | 0.72 (0.57–0.86) *** |

\*p<0.05,

\*\*p<0.01,

\*\*\*p<0.001

since COVID-19. Around 2% of overall daily tweets between 2018 and 2021 included some form of misogynistic content. This translates to millions of misogynistic tweets every day, given that around 20 billion tweets are posted daily on average. Our study draws attention to online misogyny as a topic for gender research in India, a country where internet use is on the rise and offline forms of gender-based violence are highly prevalent.

We find evidence in support of increasing trends in overall volume as well as proportion of misogynistic content on Twitter since the pandemic. However, our results show an immediate drop in the proportion of misogynistic tweets after the pandemic, followed by sustained and significant growth in the next two years. This could be because of an increase in tweets related to other relevant topics such as COVID-19, health services, and vaccines, right after the onset of the pandemic. Nonetheless, the sustained increasing trend after the initial drop is worrisome, and warrants further assessments of long-term changes. Our study contributes to the existing literature that shows significant increases in other offline forms of VAW since the pandemic.

We find that online misogyny exists in the form of sexual objectification, sexist abusive content, content that threatens to harm, asserts authority, justifies VAW, and dismisses feminism, with sexual objectification and sexual abuse of women being the most common forms. Multiple prior studies have shown online sexual harassment and victimization to be associated with adverse mental and psychological consequences, particularly for young girls [44, 45]. Online sexual harassment is unique compared to offline experiences in that individuals who are not directly attacked by the abusers/social media users, are also exposed to the sexually charged posts. In addition to causing mental distress, for young users of the Internet, exposure to such content might cause endorsements of patriarchal beliefs that view women as sexual beings and normalize VAW, leading to a continuation of the cycle of violence [46, 47]. While findings are related to the online world, this is indicative of an ongoing acceptability of sexual harassment in the country [48]. We find that sexual objectification related tweets also include content that shame women for their real or presumed sexual activity. Our findings show that online abusers often use women's sexual autonomy as a weapon for demeaning and disrespecting them, further perpetuating adverse gender norms that deny women sexual freedom.

Another manifestation of online misogyny is the dismissal of feminist efforts, and justification of VAW. Resistance to advancement of gender equality is a common feature of the feminist struggle, and like the offline world, it is evidently present in the Indian Twitter space. This

'backlash' effect has been documented for offline movements, and it can take different forms such as denial of the issue, derailing of conversation, rejection of men's responsibility, and discrediting of feminist activities [49]. Prior research has highlighted the disproportionate amount of online misogyny and violence directed at women politicians, journalists, and women engaging in feminist debate on Twitter. Our analysis did not classify tweets that were directed at specific women, but we found many posts that included words such as "*feminazis*", and dismissed feminism as a "*problem*" and a "*virus*". It is likely that such posts were directed at individuals who engage in conversations related to feminism or gender equality. Denial of the problem by focusing on fake rape allegations was another common theme across tweets categorized as justification of VAW. Such content, in addition to spreading disinformation, has the potential to widen digital gender inequalities by acting as triggering content for survivors of gender-based violence.

Currently, there are no dedicated legislations in India against gender-based cyber violence [50]. The laws protecting citizens from cyber hate do not specifically recognize sexist or misogynistic trolling. Our findings show that along with strengthening of laws and regulations to prevent online misogyny, shifting entrenched gender norms will be critical to achieving gender equality in digital spaces. Norms-based interventions addressing gender-based violence in general can consider inclusion of elements of digital violence in their programs.

In addition to providing insights on the prevalence and characteristics of online misogyny, our work is one of the first attempts to detect misogyny across a very large dataset of tweets from India. We tested multiple machine learning models, and found the pre-trained language model, BERT, to be the best performing models. However, our analysis is limited to English tweets only, and does not cover tweets in any Indian vernacular language. Future studies should consider building multilingual models to provide improved insights related to misogyny on Twitter in India.

Our study has a few additional limitations. First, our findings are likely to have an urban bias, representative of the English-speaking population, since our data was limited to English tweets only. Next, this study only includes geotagged tweets from India, which constitutes around 2% of the overall tweets. However, few prior studies have noted that the large number of tweets created daily ensure the adequateness of geotagged tweets in being a good representation of overall Twitter conversations [51, 52]. Third, we examine the changes in prevalence of misogyny before and after the start of the pandemic. We do not take into account other relevant key events that might trigger changes in the amount of misogynistic expression. Finally, our study focusses solely on the misogyny directed against women in India, and does not cover transmisogyny or different intersectional categories as relates to race, ethnicity, age, and sexual orientation. Future research should consider including these aspects, given that younger individuals, and individuals belonging to minority groups are more likely to experience violence online [5].

## Conclusions

The past couple of years have noted increasing activism against the growing levels of misogyny on Twitter as well as other social media platforms. Our findings emphasize the need for inclusion of digital violence in the broader policy discourse on gender-based violence; online misogyny is a continuum of violence experienced by women in their offline worlds in India. Aggressive and hateful tweets that target women in the country attempt to reinforce traditional gender norms, especially those relating to idealized sexual behavior and framing of women as sexual beings. Our study provides evidence on increasing trends of misogyny on Indian Twitter, and emphasizes the utility of machine learning methods in examination of aspects related

to VAW, which can support future research and development of interventions to make these digital spaces equitable and welcome for women.

## Supporting information

**S1 File. Annotation guidelines/manual for undergraduate student coders.**
(PDF)

**S2 File. Definition and example tweet for each category of online misogyny.**
(DOCX)

## Acknowledgments

We are grateful to Dr. James Fowler who supported us in accessing the Twitter data. We thank Riley Saham, Mairen Oates, Aviram Raj-Silverman, Lucas Fowler, and Max Goldberg, for serving as qualitative coders of the Twitter dataset. We are grateful to Wendy Wei Cheung for carrying out the qualitative coding of misogynist tweets.

## Author Contributions

**Conceptualization:** Nabamallika Dehingia.

**Data curation:** Nabamallika Dehingia.

**Formal analysis:** Nabamallika Dehingia.

**Funding acquisition:** Anita Raj.

**Methodology:** Nabamallika Dehingia, Julian McAuley, Lotus McDougal, Jay G. Silverman, Anita Raj.

**Supervision:** Julian McAuley, Lotus McDougal, Elizabeth Reed, Jay G. Silverman, Lianne Urada, Anita Raj.

**Writing – original draft:** Nabamallika Dehingia.

**Writing – review & editing:** Nabamallika Dehingia, Julian McAuley, Lotus McDougal, Elizabeth Reed, Jay G. Silverman, Lianne Urada, Anita Raj.

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
