## [Decision Letter · Decision Letter 0]

27 Mar 2023

PONE-D-22-30864Violence against women on Twitter in India: testing a taxonomy for online misogyny and measuring its prevalence during COVID-19PLOS ONE

Dear Dr. Dehingia,

Thank you for submitting your manuscript to PLOS ONE. After careful consideration, we feel that it has merit but does not fully meet PLOS ONE’s publication criteria as it currently stands. Therefore, we invite you to submit a revised version of the manuscript that addresses the points raised during the review process.

We look forward to receiving your revised manuscript.

Kind regards,

Mahdi Zareei

Academic Editor

PLOS ONE

Journal Requirements:

Reviewers' comments:

Reviewer's Responses to Questions

**Comments to the Author**

1. Is the manuscript technically sound, and do the data support the conclusions?

Reviewer #1: Yes

Reviewer #2: Partly

2. Has the statistical analysis been performed appropriately and rigorously? 

Reviewer #1: Yes

Reviewer #2: Yes

3. Have the authors made all data underlying the findings in their manuscript fully available?

Reviewer #1: Yes

Reviewer #2: Yes

4. Is the manuscript presented in an intelligible fashion and written in standard English?

Reviewer #1: Yes

Reviewer #2: Yes

5. Review Comments to the Author

Reviewer #1: This is an interesting article and would be of interest to readers. However the manuscript would be improved with reorganizing with the background section focusing on the problem (digital harassment/threats) globally, why focus on LMIC, specifically India, significant use of twitter vs. other LMIC, English speaking, social and gender norms that reinforce inequality, etc) consequences of digital harassment/threats on women/girls, specific details on COVID-19 in context and why important to examine the association and then move to the theory/framework and categories that guides the research and the study purpose and question. The methods section is dense, perhaps divide into phases/sections related to 1. geotagging locations of 30 million tweets (also how did you get access to the tweets for analysis), 2) Qualitative analysis of tweets for misogyny and non-misogyny (curious how students trained), 3. Test/Train models to determine best fit and 4) Analysis of misogyny and COVID in the digital space using model. The most lacking section of the paper is, what are the implications of these findings? What should/could be done to prevent/reduce digital abuse/harassment during a national/global crisis like a pandemic? What the policy and future research implications?

Reviewer #2: In general the authors show competence, for this is an interesting and well-developed study.However, I was left with some question and recommendations that I'd like to see answered in the paper itself. That said, I believe this study should be published, once these concerns are addressed.

- When describing the hypothesis and conclusions, these are worded as universal but the study is restricted to tweets from India.

- Why the authors didn't explore modern NLP classifiers/techniques? We see TF-IDF unigrams used with SVMs, bayes, logistic regression and a multi-layered perceptron. Nothing wrong with those, but there are plenty of newer architectures with better results.

- Are you doing something to mitigate the unbalanced dataset used for the supervised classification? Of the ~40K tweets used for training, only 3.8K were classified as misogynist, and of those, some categories accounted for ~1% of samples.

- My mayor concern here is that I missed a connection, and to me this work reads as two papers. First, a taxonomy for misogyny is proposed and used for the classifiers, then this classifier is used on tweets to measure said misogyny. But I missed the connecting idea between the two. If the goal is to defend the proposed taxonomy, I'd expect a comparison against previous taxonomies. If the goal is the time-series analysis, I don't know why the previous taxonomies weren't enough to make the measurements. Thus I read this work as having those two separate goals, and each has its well-earned merits, but it lacks a paragraph somewhere connecting these two ideas.

Small notes on form:

1. Line [147] the figure is missing

2. Line [311] this conclusion is not constraint to India, while the study was

3. Line [319] can we justify this with most being less than 1% of the ground truth tweets

6. PLOS authors have the option to publish the peer review history of their article (what does this mean?). If published, this will include your full peer review and any attached files.

Reviewer #1: No

Reviewer #2: No

---

## [Author Response · Author response to Decision Letter 0]

14 Jun 2023

Reviewers' comments:

Reviewer #1: 

This is an interesting article and would be of interest to readers. However the manuscript would be improved with reorganizing with the background section focusing on the problem (digital harassment/threats) globally, why focus on LMIC, specifically India, significant use of twitter vs. other LMIC, English speaking, social and gender norms that reinforce inequality, etc) consequences of digital harassment/threats on women/girls, specific details on COVID-19 in context and why important to examine the association and then move to the theory/framework and categories that guides the research and the study purpose and question. 

Response: Thank you for this feedback. We have revised the Introduction section to focus on the issue of online misogyny and its potential impacts on women’s well-being. We then highlight why studying this topic for LMICs, and particularly for India, is crucial. The first two paragraphs of the Introduction are copied below:

“Violence against women (VAW) on online platforms is a violation of women’s digital rights. It can push women out of online spaces, and impact their social, economic, as well as health outcomes [1]. One of the most prevalent forms of online VAW, online misogyny or hate speech against women refers to any content that tries to threaten, intimidate, and shame women, or any rhetoric that emphasizes the authority of men over women [1]. According to a multi-country online survey in 2019, around two-thirds of women users of the internet reported to have received some form of sexist or hateful language designed to attack or humiliate them [2]. Such experiences can often cause increased anxiety, depression, and lower self-esteem among women [3, 4]. The extent and frequency of misogynistic attacks varies across different groups of women; younger women, and women belonging to marginalized racial or sexual identity groups are more at risk of experiencing online misogyny [2, 5]. The past two years have thus noted increasing amount of calls globally, for hate speech regulation on social media platforms [10, 11]. 

Majority of the existing research on online violence, including online misogyny, has focused on high-income countries, primarily due to greater digital access and use of social media platforms in these regions [12]. To our knowledge, no academic study has examined the characteristics of online misogyny in low- and middle-income countries (LMIC). This presents a key gap in literature, given that the use of social media, and consequently, the amount of online hateful content, has increased rapidly in LMICs in recent years [13, 14]. India, in particular, is an important geography to examine online misogyny. Compared to many other LMICs, adverse gender norms and offline forms of gender-based violence are more prevalent in the country [15, 16], putting women at a higher risk of experiencing violence in online spaces. Use of social media platforms is also on the rise in the country. As of January 2022, India had over 25 million active Twitter users, with a 30% increase in average daily users in 2020. Although Twitter has an urban bias; most Twitter users in the country are likely to be educated, living in cities, and of middle to high wealth status [17]. Nonetheless, Twitter is being frequently used as a key platform for feminist discussions and organizing in India [18], which can increase the likelihood of misogynist expression as a backlash [19]. The current study aims to contribute to the growing literature on online VAW on social media platforms, by measuring online misogyny on Twitter in India.”

The methods section is dense, perhaps divide into phases/sections related to 1. geotagging locations of 30 million tweets (also how did you get access to the tweets for analysis), 2) Qualitative analysis of tweets for misogyny and non-misogyny (curious how students trained), 3. Test/Train models to determine best fit and 4) Analysis of misogyny and COVID in the digital space using model. 

Response: Thank you for this feedback. We have organized the Methods section as suggested, into smaller sub-sections. 

The undergraduate students who were responsible for the qualitative coding were provided with multiple sessions of instruction on the definition of online misogyny, its different categories, and potential challenges in classification of tweets as misogyny/non-misogyny. They were also provided with a manual, which included detailed information on how to carry out the coding, with multiple examples of tweet classification. We have now included the manual as supplementary information in the submission.

The most lacking section of the paper is, what are the implications of these findings? What should/could be done to prevent/reduce digital abuse/harassment during a national/global crisis like a pandemic? What the policy and future research implications?

Response: Thank you. We appreciate this feedback. We have revised the Discussion section to include implications of this work. We note the need for strengthening current laws and regulations preventing online gender-based violence in India. Currently, there are no dedicated legislations in India against gender-based cyber violence. The laws protecting citizens from cyber hate do not specifically recognize sexist or misogynistic trolling. We also include a brief discussion on the potential of norms-based interventions in addressing online misogyny. Our findings indicate that online misogyny is a continuum of violence experienced by women in their offline worlds. As such, shifting entrenched gender norms will be critical to achieving gender equality in digital spaces.

We also highlight that given the current lack of studies on online misogyny, there is a need for generating more knowledge and evidence around this topic and its risk factors, for effective policy and intervention development. Our research demonstrates the use of publicly available social media data and machine learning models as effective analytical tools to study this topic. Our analysis however is limited to English tweets only, and does not cover tweets in any Indian vernacular language. Future studies should consider building multilingual models to provide improved insights related to misogyny on Twitter in India.

Reviewer #2: 

In general the authors show competence, for this is an interesting and well-developed study. However, I was left with some question and recommendations that I'd like to see answered in the paper itself. That said, I believe this study should be published, once these concerns are addressed.

- When describing the hypothesis and conclusions, these are worded as universal but the study is restricted to tweets from India.

Response: Thank you for this feedback. We have revised the manuscript to clarify that our analysis as well as findings are specific to India, and are not universal.

- Why the authors didn't explore modern NLP classifiers/techniques? We see TF-IDF unigrams used with SVMs, bayes, logistic regression and a multi-layered perceptron. Nothing wrong with those, but there are plenty of newer architectures with better results.

Response: Thank you. We appreciate this feedback. We ran the entire analysis with the language model BERT, Bidirectional Encoder Representations from Transformers. BERT is a pre-trained language model (BERT is pretrained on a large corpus of English texts from Wikipedia and BookCorpus), with transformer architecture. Transformers learn contextual relations between words; they are deep learning models where each output element is connected to every input element and the weightings between them are dynamically calculated. While previous models examine text sequences in a single direction (left-to-right or right-to-left training), BERT’s key difference and innovation is the application of bidirectional training. The BERT Transformer encoder reads the entire sequence of words at once. Specifically, we used the bert-large-uncased version, which consists of 24 layers (1024 hidden dimensions), 16 attention heads, and a total of 340M parameters. The Transformers library in Python was used to implement this approach.

The BERT model performed significantly better than the other supervised classification models in predicting misogyny, with an AUC of 0.91 and F-1 score of 0.84. Our overall results however did not change in terms of increasing prevalence of misogyny post COVID-19.

-Are you doing something to mitigate the unbalanced dataset used for the supervised classification? Of the ~40K tweets used for training, only 3.8K were classified as misogynist, and of those, some categories accounted for ~1% of samples.

Response: Thank you. We appreciate this feedback. We ran the analysis by using the oversampling method, to address the unbalanced dataset. We used the RandomOverSampler package from the sci-kit library in Python to oversample. However, this did not lead to any improvements in evaluation metrices for our best performing model, the BERT model.

- My mayor concern here is that I missed a connection, and to me this work reads as two papers. First, a taxonomy for misogyny is proposed and used for the classifiers, then this classifier is used on tweets to measure said misogyny. But I missed the connecting idea between the two. If the goal is to defend the proposed taxonomy, I'd expect a comparison against previous taxonomies. If the goal is the time-series analysis, I don't know why the previous taxonomies weren't enough to make the measurements. Thus I read this work as having those two separate goals, and each has its well-earned merits, but it lacks a paragraph somewhere connecting these two ideas.

Response: We appreciate this comment. The primary goal of this work was to assess whether the prevalence of online misogyny increased since the pandemic. However, upon reviewing the existing literature on measurement of online misogyny, we observed the lack of a standard taxonomy, and the few existing studies on this topic were from high-income country contexts. We believe it was important to adapt these previously tested taxonomies to the context of India. Based on our preliminary work on online misogyny detection across countries in South Asia (including India), and existing literature on online misogyny as well as overall gender-based violence in the Indian context, we added an extra dimension of ‘dismissal of feminist efforts’ to the existing taxonomy of online misogyny. We elaborate this rationale in the manuscript (Introduction and Methods section).

We recognize that the adaptation and testing of the taxonomy could have been a separate research paper, but our project commitments require us to produce one manuscript on the relationship between COVID-19 and online misogyny. We have revised the manuscript to clarify the reason for adapting and testing the taxonomy for misogyny prior to its prevalence measurement.

Small notes on form:

1. Line [147] the figure is missing

Response: The PlosOne guidelines for figures requires that only the figure caption is included in the manuscript. The figure has been provided as a separate file.

2. Line [311] this conclusion is not constraint to India, while the study was

Response: Corrected.

3. Line [319] can we justify this with most being less than 1% of the ground truth tweets

Response: While the proportion of misogynistic tweets is low, the absolute volume of misogynistic tweets is worrying, given that billions of tweets are generated daily. We argue that the presence of such hostile content is an issue that requires policy attention, to ensure that digital spaces are safe for women. We include this discussion in the text.

---

## [Decision Letter · Decision Letter 1]

19 Jul 2023

PONE-D-22-30864R1Violence against women on Twitter in India: testing a taxonomy for online misogyny and measuring its prevalence during COVID-19PLOS ONE

Dear Dr. Dehingia,

Thank you for submitting your manuscript to PLOS ONE. After careful consideration, we feel that it has merit but does not fully meet PLOS ONE’s publication criteria as it currently stands. Therefore, we invite you to submit a revised version of the manuscript that addresses the points raised during the review process.

 Please see the request under "journal requirements" below.

We look forward to receiving your revised manuscript.

Kind regards,

Mahdi Zareei

Academic Editor

PLOS ONE

Journal Requirements:

We note that your manuscript, in particular Table 1, contains terms typically considered offensive that would also be considered relevant to the study and necessary to meet PLOS ONE publication criteria for reporting and reproducibility.We request that you please move Table 1 to Supporting Information due to its graphic nature, and please include the following text at the beginning of the file:

"Disclaimer: This file includes words or language that is considered profane, vulgar or offensive by some readers. Due to the topic studied in this article, quoting offensive language is academically justified but we nor PLOS in no way endorse the use of these words or the content of  the quotes."

Additionally, please review your reference list to ensure that it is complete and correct. If you have cited papers that have been retracted, please include the rationale for doing so in the manuscript text, or remove these references and replace them with relevant current references. Any changes to the reference list should be mentioned in the rebuttal letter that accompanies your revised manuscript. If you need to cite a retracted article, indicate the article’s retracted status in the References list and also include a citation and full reference for the retraction notice.

Reviewers' comments:

Reviewer's Responses to Questions

**Comments to the Author**

1. If the authors have adequately addressed your comments raised in a previous round of review and you feel that this manuscript is now acceptable for publication, you may indicate that here to bypass the “Comments to the Author” section, enter your conflict of interest statement in the “Confidential to Editor” section, and submit your "Accept" recommendation.

Reviewer #1: All comments have been addressed

Reviewer #2: All comments have been addressed

2. Is the manuscript technically sound, and do the data support the conclusions?

Reviewer #1: Yes

Reviewer #2: Yes

3. Has the statistical analysis been performed appropriately and rigorously? 

Reviewer #1: Yes

Reviewer #2: Yes

4. Have the authors made all data underlying the findings in their manuscript fully available?

Reviewer #1: Yes

Reviewer #2: Yes

5. Is the manuscript presented in an intelligible fashion and written in standard English?

Reviewer #1: Yes

Reviewer #2: Yes

6. Review Comments to the Author

Reviewer #1: The authors have addressed my previous comments and that of the other reviewer. I think the manuscript is of the quality to be published in PLOS ONE

Reviewer #2: (No Response)

7. PLOS authors have the option to publish the peer review history of their article (what does this mean?). If published, this will include your full peer review and any attached files.

Reviewer #1: No

Reviewer #2: No

---

## [Author Response · Author response to Decision Letter 1]

1 Sep 2023

Thank you for the review. We have moved Table 1 to Supporting Information. We have also removed abusive words from the manuscript text. These words were included as examples of selected tweets.

---

## [Editor Report · Decision Letter 2]

13 Sep 2023

Violence against women on Twitter in India: testing a taxonomy for online misogyny and measuring its prevalence during COVID-19

PONE-D-22-30864R2

Dear Dr. Dehingia,

We’re pleased to inform you that your manuscript has been judged scientifically suitable for publication and will be formally accepted for publication once it meets all outstanding technical requirements.

Kind regards,

Mahdi Zareei

Academic Editor

PLOS ONE
---

## [Editor Report · Acceptance letter]

25 Sep 2023

PONE-D-22-30864R2 

Violence against women on Twitter in India: testing a taxonomy for online misogyny and measuring its prevalence during COVID-19 

Dear Dr. Dehingia:

I'm pleased to inform you that your manuscript has been deemed suitable for publication in PLOS ONE. Congratulations! Your manuscript is now with our production department. 

Kind regards, 

on behalf of

Dr. Mahdi Zareei 

Academic Editor

PLOS ONE